# Contribution of tree community structure to forest productivity across a thermal gradient in eastern Asia

Tetsuo I. Kohyama [1,2,3,17] ✉, Douglas Sheil [4,5,6,17], I-Fang Sun[7], Kaoru Niiyama[8], Eizi Suzuki[9], Tsutom Hiura[3], Naoyuki Nishimura [10], Kazuhiko Hoshizaki [11], Shu-Hui Wu[12], Wei-Chun Chao[13], Zamah S. Nur Hajar[14], Joeni S. Rahajoe[15] & Takashi S. Kohyama [1,16,17]

Despite their fundamental importance the links between forest productivity, diversity and climate remain contentious. We consider whether variation in productivity across climates reflects adjustment among tree species and individuals, or changes in tree community structure. We analysed data from 60 plots of humid old-growth forests spanning mean annual temperatures (MAT) from 2.0 to 26.6 °C. Comparing forests at equivalent aboveground biomass (160 Mg C ha⁻¹), tropical forests ≥24 °C MAT averaged more than double the aboveground woody productivity of forests <12 °C ($3.7 \pm 0.3$ versus $1.6 \pm 0.1$ Mg C ha⁻¹ yr⁻¹). Nonetheless, species with similar standing biomass and maximum stature had similar productivity across plots regardless of temperature. We find that differences in the relative contribution of smaller- and larger-biomass species explained 86% of the observed productivity differences. Species-rich tropical forests are more productive than other forests due to the high relative productivity of many short-stature, small-biomass species.

Forest ecosystems play a central role in global carbon dynamics due to their biomass and productivity[1,2]. Nevertheless, we remain uncertain how forest productivity and other fundamental ecosystem functions are related to tree species diversity[3]. Multiple factors, including stand structure[4–6], spatial scales[7–9], and species composition[10–12] have been suggested to explain productivity-diversity relationships. Indeed, whether stand level (i.e., local scale) primary productivity reflects climate, edaphic characteristics, and/or tree species diversity, and the mechanisms, remain debated[13–16].

Cross-forest comparisons and analyses suffer various limitations. Standard estimates of net primary productivity from plot census data (i.e., reflecting tree growth and recruitment, hereafter, "woody productivity" for short[6,9]) neglect production by trees that die between censuses[17]. This neglect coupled with among-population

[1]Faculty of Environmental Earth Science, Hokkaido University, Sapporo 060-0810, Japan. [2]Center for Far Eastern Studies, University of Toyama, Toyama 930-8555, Japan. [3]Department of Ecosystem Studies, The University of Tokyo, Tokyo 113-8657, Japan. [4]Department of Environmental Sciences, Wageningen University & Research, Wageningen, The Netherlands. [5]Center for International Forestry Research, Kota Bogor, Jawa Barat 16115, Indonesia. [6]Faculty of Environmental Sciences and Natural Resource Management, Norwegian University of Life Sciences, Ås, Norway. [7]Center for Interdisciplinary Research on Ecology and Sustainability, National Dong Hwa University, Hualien 974301, Taiwan. [8]Department of Forest Vegetation, Forest and Forest Products Research Institute, Tsukuba 305-8687, Japan. [9]Research Center for the Pacific Islands, Kagoshima University, Kagoshima 890-8580, Japan. [10]Faculty of Informatics, Gunma University, Maebashi 371–8510, Japan. [11]Department of Biological Environment, Akita Prefectural University, Akita 010-0195, Japan. [12]Taiwan Forestry Research Institute, Taipei 100060, Taiwan. [13]Department of Forestry and Natural Resources, National Chiayi University, Chiayi City 600355, Taiwan. [14]Forestry and Environment Division, Forest Research Institute Malaysia, Kepong, Selangor 52109, Malaysia. [15]Research Center for Ecology and Ethnobiology, National Research and Innovation Agency, Cibinong, Jawa Barat 16911, Indonesia. [16]Center for Southeast Asian Studies, Kyoto University, Kyoto 606-8501, Japan. [17]These authors contributed equally: Tetsuo I. Kohyama, Douglas Sheil, Takashi S. Kohyama. ✉e-mail: tetsuo.kohyama@gmail.com

heterogeneity in turnover rates causes biases that impair confident comparisons of estimates among sites[18,19]. To reduce such biases, we devised and applied improved formulae for population woody productivity[20], in which stand-level productivity is estimated from the sum of the productivities of the component populations. Our previous application of these methods to a diverse Malaysian rain forest revealed how species-level relative (i.e., per-biomass) woody productivity decreases with species standing biomass (per area)[21]. We also found that for each species, its per-area standing biomass ("species biomass") is better predicted by maximum tree mass (size) than by abundance (counts per unit area)[21]. Based on these patterns, we proposed that the relative contributions of smaller and larger species—i.e., "community structure"—may influence how forest productivity relates to tree species composition, richness, and climate.

Based on species-level productivity-biomass relationships, we propose two alternative hypotheses for differences in productivity when considering a forest stand with a fixed-standing biomass. Our species-response hypothesis (Fig. 1a) posits that most trees and species populations share a tendency to increase productivity in warmer climates, which collectively leads to higher stand productivity[22]. This hypothesis implies that differences in composition and community structure seen in forests under different temperatures play little role. In contrast, our community structure hypothesis (Fig. 1b) posits that the nature of the tree community—in terms of the frequency distribution of species standing biomass and species richness—determines stand-level differences in productivity in forests from different climates. Both tree species richness and the proportion of species possessing higher relative productivity (predominantly species with shorter stature and lower overall biomass per unit area)[21] tend to increase with increasing warmth[23–25] leading such forests to support higher total stand productivity. Figure 1 presents simulated examples from these two hypotheses assuming inter-specific productivity-biomass power-law models. Each hypothesis can explain how productivity changes with climate.

Here we evaluate each hypothesis, by examining repeated tree censuses from 60 1-ha primary or old-growth forest plots across insular eastern Asia, ranging 6.8°S–44.4°N latitude, 2.0–26.6 °C in mean annual temperature (MAT) and 998–5791 mm yr⁻¹ in mean annual precipitation (AP) (Supplementary Fig. 1, Supplementary Data 1, 2). Water stress is low across all these sites. Available data indicates that all the locations have an excess of water availability in terms of AP minus annual potential evapotranspiration (AP-PET) of 59 to 5049 mm yr⁻¹, and the climatic moisture types[26] include 'perhumid' (48 plots), 'humid' (11 plots) or 'moist sub-humid' (1 plot) (Methods, Supplementary Data 2). The entire region is generally considered fertile due to volcanic activity and continental dust transportation[27]. Nonetheless, several of our plots are located in specific substrates (tropical heath, peat-swamp, and limestone) (Supplementary Data 1), which showed no significant differences from other plots in our analyses.

## Results

### Species-level productivity

Within each of the 60 forest plots, the relative aboveground woody productivity of each species (i.e., species woody productivity per-species biomass) showed a similar pattern across plots. The values decreased with species' standing aboveground biomass, in a similar manner regardless of plot and location, with a power-law exponent of $-0.14 \pm 0.01$ (Fig. 2a). For each species its biomass per unit area was also strongly correlated with the maximum tree mass observed for that species (log-log model, $R^2 = 0.85$) and with its maximum tree height where these data are available ($R^2 = 0.85$, for 11 plots) (Supplementary Fig. 2). Likewise, the relative woody productivity of each species also decreased similarly when plotted against that species' maximum tree mass with a power-law exponent of $-0.18 \pm 0.01$ regardless of plot and climate (Fig. 2b). Furthermore, the form of the scatters revealed by plotting relative growth rates of individual trees against tree mass were similar across plots and biomes (Supplementary Fig. 3), the scatters revealed by plotting species absolute productivity against species per-area biomass (Supplementary Fig. 4).

The relative contribution of small-biomass species varied considerably among plots and biomes. Species represented by low biomass ("small-biomass species" having less than 3 Mg C ha⁻¹) provided 37, 16, and 5% of forest productivity in tropical, cool-temperate, and sub-boreal forests respectively (Fig. 2c). The marked difference in the relative contribution of these small-biomass species support our community structure hypothesis.

### Forest structure and temperature

Plot-level tree species richness in log scale (ln SR) showed a clear positive correlation with mean annual temperature, MAT (semi-log model, $R^2 = 0.64$; Fig. 3a). In contrast, forest aboveground biomass was weakly and positively correlated with both MAT ($R^2 = 0.14$ by semi-log

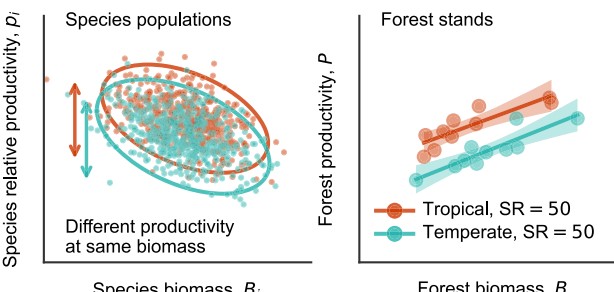

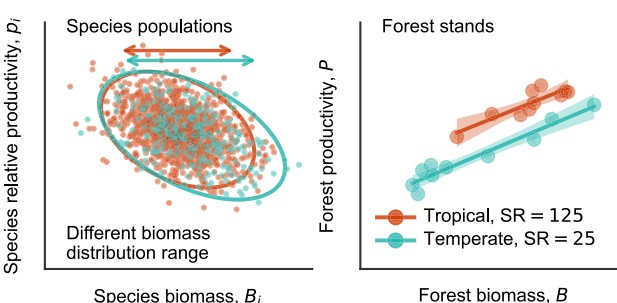

**Fig. 1 | Two hypotheses explaining higher forest woody productivity in warmer forest stands in relation to inter-specific productivity-biomass relationships.** **a** The species response hypothesis assumes that tree species possess distinct productivity-biomass power-law relationships depending on temperature such that relative aboveground woody productivity $p_i$ of species $i$ tends to be higher in tropical versus temperate forests at the same species' standing aboveground biomass $B_i$, leaving that frequency distribution of species biomass and per-stand species richness (SR) of each stand are the same among biomes. **b** The community structure hypothesis assumes that species possess similar productivity-biomass relationships regardless of temperature, i.e., species relative woody productivity $p_i$ is not different between tropical and temperate species at the same species aboveground biomass $B_i$, while SR is larger and mean species biomass is smaller in tropical forest stands. Based on synthetic data generated assuming a bivariate normal distribution of ln $p_i$ and ln $B_i$, with a common correlation slope (or, power-law exponent) of $-0.15$. Other coefficient values for generating random data are in Methods. Each of 10 stands in each forest has SR shown in right-hand panels. In the left-hand panels, the 95% prediction ellipses are shown in inter-specific $p_i$-$B_i$ relationships. The right-hand panels show species-aggregated, stand-level woody productivity $P = \Sigma_i p_i B_i$ and stand biomass $B = \Sigma_i B_i$, with predicted means and 95% confidence intervals of the power-law model fitting. All axes are on log scale. Source data are provided as a Source Data file.

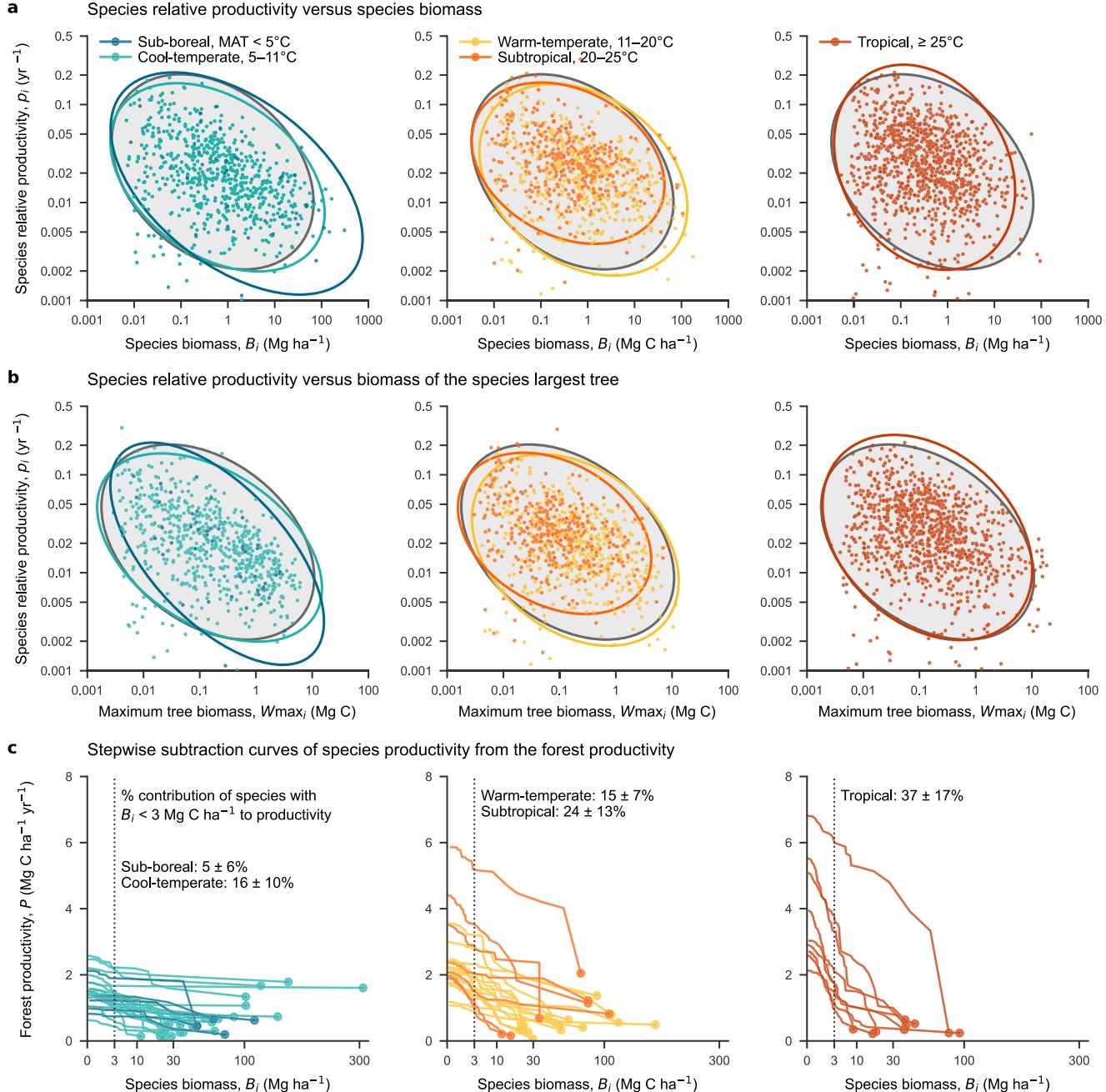

**Fig. 2 | Species-level standing biomass and woody productivity across 2604 species populations (excluding rare aggregated species populations) in 60 forest plots in eastern Asia.** Plots are grouped by mean annual temperature (MAT). **a** Relationships between per-plot species-$i$ relative aboveground woody productivity $p_i$, against species aboveground biomass $B_i$, and **b** those between species relative productivity against species' maximum tree mass, $W\max_i$, both **a** and **b** on log–log scale. The 95% log-normal prediction ellipses are shown for each of five biomes grouped by mean annual temperature (MAT) by corresponding colours, and the common ellipse for all species populations in all plots by filled grey. Fitted power-law models were: $p_i = a_{plot} B_i^{-0.14\pm0.01}$ with $a_{plot}$ ranging [0.011, 0.058], and $p_i = a'_{plot} W\max_i^{-0.18\pm0.01}$ with $a'_{plot}$ ranging [0.0078, 0.043]. **c** Decrease of forest-plot woody productivity $P = \Sigma_i p_i B_i$ with stepwise reduction of species biomass $B_i$ from the smallest to the largest, in which lines show each plot, and the circles at the right show species woody productivity $P_i = p_i B_i$ of the largest biomass species. Mean and s.d. percentage of the productivity of small-biomass species ($B_i < 3$ Mg C ha$^{-1}$, dashed line) to forest productivity $P$ are presented; $x$ axis is on square-root scale and $y$ axis on normal scale. Source data are provided as a Source Data file.

model; Fig. 3b), and species richness ($R^2 = 0.09$ by log-log model; Supplementary Fig. 5). Reflecting the increase in tree species richness being greater (ranging 5–365 ha$^{-1}$) than that in stand biomass (ranging 77–343 Mg C ha$^{-1}$) across our sampled gradient, average per-species biomass per plot decreases as MAT increases ($R^2 = 0.34$ by semi-log model; Fig. 3c). Furthermore, the plot-level standard deviation of species biomass also decreases with MAT ($R^2 = 0.20$ by semi-log model; Fig. 3d), indicating larger interspecific variation in forests from cooler

versus warmer climates. Forest biomass and tree species richness are only weakly correlated to annual precipitation AP ($R^2 = 0.09$ and 0.07, respectively), and they appear unrelated to AP-PET (Supplementary Fig. 5).

## Forest-level productivity
We obtain the forest aboveground woody productivity of each plot, $P$ (Mg C ha$^{-1}$ yr$^{-1}$) by summing species absolute woody productivities

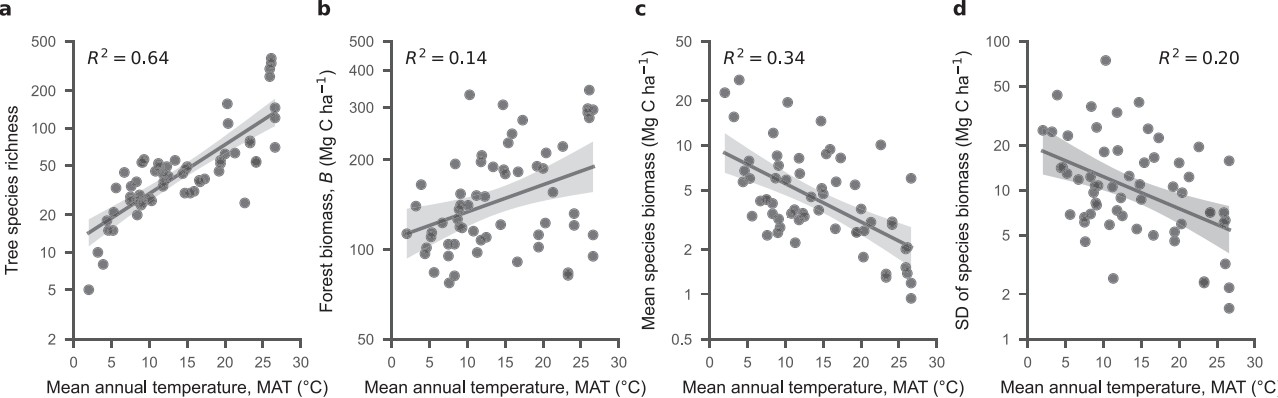

**Fig. 3 | Forest structural measures in relation to mean annual temperature (MAT) among 60 plots in eastern Asia. a** Tree species richness (i.e., species count per plot); **b** aboveground forest biomass $B$; **c** mean species aboveground biomass; **d** standard deviation of species aboveground biomass. The regression line in each panel indicate mean and 95% confidence intervals predicted by the semi-log model. The $R^2$ values of the models are shown. Forest structure measures on y axes are on a log scale. Source data are provided as a Source Data file.

$(\Sigma_i \, p_i \, B_i)^{20}$. Variation in forest woody productivity $P$ among plots was well approximated by the power-law of forest standing biomass $B$ (Mg C ha$^{-1}$) and by exponential increase with MAT (°C) at reference $B$ (namely the original data model, $R^2 = 0.62$; Fig. 4a), reflecting only weak collinearity between ln $B$ and MAT (Fig. 3b). In contrast, because of the strong collinearity between log species richness and MAT (Fig. 3a), this model sufficiently explained forest productivity without the separate inclusion of species richness. Neither of annual precipitation (AP) nor annual precipitation minus potential evapotranspiration (AP-PET) made additional contributions to the forest productivity model.

To control for differences in biomass, we estimated aboveground woody productivity (mean ± s.d.) at aboveground biomass of 160 Mg C ha$^{-1}$ for tropical (≥24 °C MAT), subtropical/warm-temperate (12–24 °C) and cool-temperate/sub-boreal (<12 °C) forests was respectively 3.7 ± 0.3, 2.5 ± 0.1 and 1.6 ± 0.1 Mg C ha$^{-1}$ yr$^{-1}$ (Fig. 4d). The estimated coefficient of dependence on MAT was 0.048 ± 0.006 °C$^{-1}$ (i.e., $P$ at the same $B$ doubles as MAT increase by (ln 2)/0.048 = 14 °C).

To evaluate how these values conform to our species response hypothesis, we generated simulated data of forest woody productivity, $P_{specRes}$ We did this assuming that every species population responded similarly to MAT employing our derived plot-specific species productivity-biomass power-law models and that every plot had a common distribution of biomass among species (see Methods). The resulting $P_{specRes}$ indicated a positive dependence on MAT (0.018 ± 0.006 °C$^{-1}$; i.e., $P$ at the same $B$ doubles as MAT increase by (ln 2)/0.018 = 39 °C) lower than seen in the original data (Fig. 4b, d). To test the alternative community structure hypothesis, we generated a comparable dataset through an approach in which we eliminated species' dependence on plot identity. We did this by swapping the productivity value of each species with that of another species with similar biomass (i.e., biomass per unit area) among the 60 plots (Methods; Supplementary Fig. 6). The resulting estimate of forest woody productivity using this replaced data, $P_{commStr}$, revealed a pattern of response to forest biomass and MAT similar to that observed in the original data (Fig. 4c, d), though the dependence on MAT was somewhat lower (0.028 ± 0.005 °C$^{-1}$; $P$ doubles by 25 °C). Relative contributions of the two hypotheses estimated by fitting regression models to original data (cf. Methods) were 24% for the species response hypothesis, and 86% for the community structure hypothesis (The sum is larger than 100%, because the simulated data $P_{specRes}$ assume that species richness is proportional to plot biomass and differs across plots).

## Discussion

We found that the inter-specific trade-off between relative woody productivity and biomass, or maximum stature[21], are common and similar among forests over an extended thermal gradient in eastern Asia (Fig. 2a, b). This inter-specific trade-off reflects the slower turnover of larger compared to smaller tree species[6,21]. While larger trees may capture more solar energy per unit of foliage due to their greater stature[28], they also possess more overall biomass per unit of leaf mass compared to smaller trees[29]. Thereby, taller trees and large-biomass species generally show lower relative productivity than smaller, shorter tree species that tend to have lower biomass per unit area. Why are these patterns consistent across climates? We presume that adaptation and acclimation to local climate within these old-growth forests ensure similar woody productivity for species with similar standing biomass and stature regardless of temperature[30–34].

In contrast to the similarity in how much species of comparable size and biomass contribute to production among biomes, we found obvious differences in the resulting stand-level woody productivity among the actual forests with markedly higher production in warmer versus cooler locations (Fig. 4). Our evaluation indicates that these differences primarily reflect variation in tree community structure, and the direct influence of temperature on individual trees and populations playing a minor role (Fig. 4, Supplementary Fig. 3).

Our plot-level data confirm that forest woody productivity was, as seen in many other studies[3,14–16], positively associated with tree species richness. Furthermore, our examination of the species-level contribution to forest-level productivity indicates the major influence of tree community structure on forest productivity. We suggest the productivity-diversity association across forest biomes is primarily indirect such that both higher tree species richness and higher stand productivity reflect increasing abundance of smaller (versus larger) stature species. Species-rich tropical forests generally comprise a greater proportion of short-stature species, and trees of such species (Fig. 3), when compared to lower diversity temperate forests[23–25]. The greater abundance of these small-biomass, short-stature species in tropical climate with higher temperatures presumably reflect differences in light penetration, enhancing the persistence and viability of understorey short-stature species along a latitudinal gradient[35]. This enhanced persistence and viability offers more space for such species which combined with greater climatic stability and enhanced speciation leads to a greater diversity of shorter-stature, higher-turnover taxa in the tropics[36–38].

The fundamental nature of the relationships indicated by our study, whether they apply in water or nutrient-limited environments, and their implications in a changing climate remain important

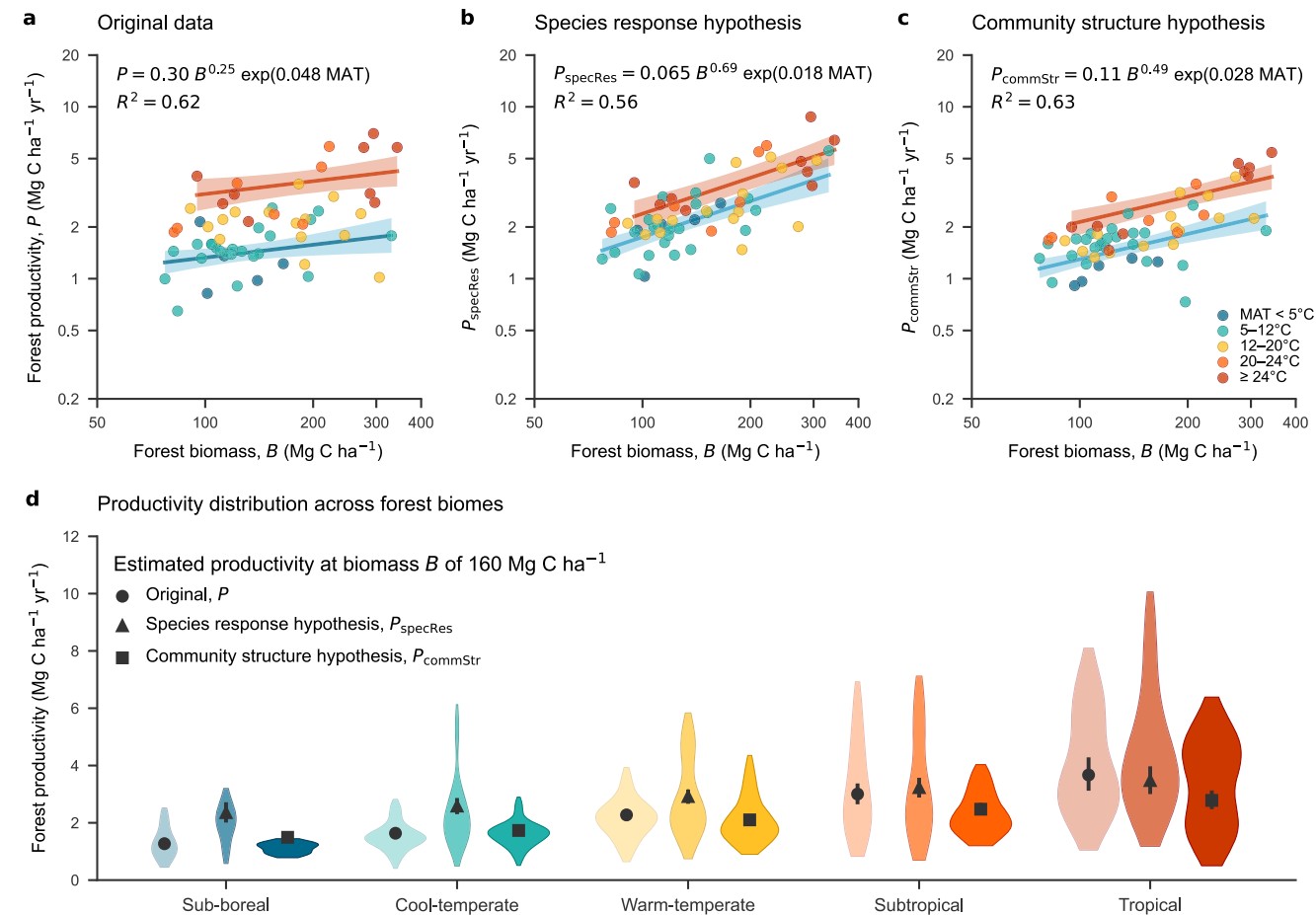

**Fig. 4 | Estimated forest-level woody productivity in relation to standing biomass and temperature among 60 plots in eastern Asia. a** Forest-level aboveground woody productivity estimated from original data $P$ (i.e., estimates based on constituent species productivity) against forest aboveground biomass per area. **b** Estimated woody productivity representing the species response hypothesis, $P_{specRes}$, against forest biomass; $P_{specRes}$ is obtained assuming that all species populations across 60 plots respond to the species productivity-biomass power-law specific to each plot of interest, and that frequency distribution of species biomass of the plot is the same as the distribution of all species populations of all plots. **c** Estimated woody productivity representing the community structure hypothesis, $P_{commStr}$, against forest biomass, where inferred species woody productivity is resampled from a species population with similar biomass drawn from all the plots. In **a–c**, regression lines indicate mean with 95% confidence intervals are shown for tropical ($\geq$24 °C, red line) versus cool-temperate/sub-boreal forests (<12 °C, blue line) predicted by power-law models. Estimates and coefficient of determination for the power-law model are shown in each panel. **d** Forest-level woody productivity distribution in five biomes grouped by mean annual temperature (MAT): coloured areas show the distribution of woody productivity with kernel density estimation; inside symbols with bars are mean and 95% confidence intervals of predicted productivity at the forest biomass of 160 Mg C ha$^{-1}$ (mean across 60 plots). Source data are provided as a Source Data file.

questions for future evaluation. While our results cast a novel light on forest productivity and biodiversity, they also open new avenues for exploration.

## Methods

### Synthetic data for Fig. 1
To provide examples of the proposed two hypotheses, i.e., species-response hypothesis and community structure hypothesis, for Fig. 1, we generated synthetic data assuming bivariate lognormal distributions of species relative woody productivity $p_i$ and species standing biomass $B_i$, where $i$ for species identity, with log-log linear, (or power-law) correlations, $\ln p_i = k + b \ln B_i$, as in left-hand panels of Fig. 1. The slope (scaling exponent) $b$ is common at −0.15, and the constant $k = −3.4$ and −3.8 for tropical and temperate forests respectively for species response hypothesis (Fig. 1a), whereas $k = −3.6$ for both 'tropical' and 'temperate' forests for the community structure hypothesis (Fig. 1b). Mean $\ln B_i$ are −0.6 for two forests in Fig. 1a, while they are −1.0 and −0.2 for tropical and temperate forest respectively in Fig. 1b, Standard deviations of $\ln B_i$ and $\ln p_i$ are 2.0 and 0.65 respectively for all forests, except those in

tropical forest in Fig. 1b are 1.6 and 0.6, respectively. In the left-hand panels, the $B_i$ axis ranges 0.005–500 (Mg C ha$^{-1}$), and the $p_i$ axis ranges 0.001–0.5 (yr$^{-1}$). In the right-hand panels, the axis for $B = \Sigma_i B_i$ ranges 50–500 (Mg C ha$^{-1}$) and the axis for $P = \Sigma_i p_i B_i$ ranges 0.5–20 (Mg C ha$^{-1}$ yr$^{-1}$).

### Forest plot data
We selected 60 forest plots located in old-growth forests along the climatic gradient of insular eastern Asia, located on Java (3 plots), Kalimantan (5 plots), Peninsular Malaysia (2 plots), Taiwan (6 plots), and the Japanese archipelago (44 plots), ranging from 6.8°S to 44.4°N latitude and from 20 to 1,880 m in elevation (Supplementary Fig. 1, Supplementary Data 1). We collected climate data for all the plots for the period 1981–2010 from CHELSA version 2.1[39]; these are the period-average annual and monthly ground surface mean temperature, precipitation, and potential evapotranspiration. The potential evapotranspiration was estimated by Hargreaves-Samani equation[40] based on monthly data of these climatic variables. Supplementary Data 2 presents mean annual temperature (MAT, °C), annual precipitation (AP, mm yr$^{-1}$), annual potential evapotranspiration (PET, mm yr$^{-1}$),

monthly-data-based Thornthwaite moisture index (TMI) and the climatic types defined by TMI[26]. The target region is in Asian monsoon climate[41,42], and moist forest ecosystems predominate from tropics in Southeast Asia to sub-boreal biomes in northern Japan. Across 60 plots, MAT ranges from 2.0 °C to 26.6 °C, AP-PET ranges from 58.5 to 5049 mm yr$^{-1}$, and plots are classified as "perhumid" or "humid" by TMI (Supplementary Data 2); the smallest TMI for the plot in cloud forest on Hahajima Island, oceanic Ogasawara Islands, where AP-PET was +217 mm yr$^{-1}$ (against +58.5 by CHELSA[39]) based on the weather station records on the island. AP-PET sowed no correlation with MAT or with any forest structural or dynamic variable, in contrast to MAT exhibiting significant correlations to all forest variables (Supplementary Fig. 5). We therefore mainly employ MAT to quantify climatic dependence of the 60 plots. According to bioclimatic classification of the region[43,44], we define forest biomes into tropical (MAT ≥ 24 °C), subtropical (20–24 °C), warm-temperate (12–20 °C), cool-temperate (5–12 °C) and sub-boreal or subalpine (<5 °C). Dominant tree life forms are evergreen broad-leaved in tropical, subtropical, and warm-temperate forests, deciduous broad-leaved in cool-temperate, and evergreen coniferous in sub-boreal forests (Supplementary Data 3). We employed biomes to approximate forests into groups, while we used MAT values of each plot to analyse temperature dependence of forest stands and species populations.

As in Supplementary Data 1, 60 plots are ~1 ha in horizontal area (56 plots are exactly 1 ha, and others are 0.95, 0.98, 1.04, and 1.05 ha). Data include 41 (out of 44) plots across Japanese archipelago (code a in Supplementary Data 1; the original data is available at https://www.biodic.go.jp/moni1000/findings/data/)[45]. Two cool-temperate mixed broadleaf-conifer forest plots (code c)[20], subalpine spruce-fir forest (code h, raw data originally uploaded here by T. S. Kohyama), two lowland mixed dipterocarp forest plots in West Kalimantan (code j)[46], three lowland heath or peat-swamp forests in Central Kalimantan (code f)[47–49], three tropical montane forest plots in West Java (code d)[50], two lowland mixed dipterocarp forest plots in Peninsular Malaysia (code i)[51,52], two subtropical karst forest plots (codes c)[53], two subtropical montane forest plots (code d)[54], and two warm-temperate montane forests in Taiwan[55,56]. Where we have large continuous forest plots (codes b, c, g, i, h), we sampled every two 1-ha plots most distant from the other.

In each plot, tree censuses of stem diameter at breast height at 1.3 m above ground, or a marked position above buttress, were carried out for all stems ≥5 cm diameter appeared in either of two censuses with interval of ~5 years (ranging 2.5 and 8.5 years). We estimated individual-tree total aboveground carbon mass and leaf carbon mass from stem diameter using local allometric equations if available; otherwise, we used the generic allometric equations for eastern Asia[57] with species-specific wood densities[58,59]. Generic estimates capture 89–99% of site-specific measures of tree mass[57]. We used genus or family averages of wood density when species-specific data were not available, and when even these were not available, we used lifeform-specific equations (evergreen broadleaf, deciduous broadleaf, and evergreen conifer). We used the factor of 0.5 to convert oven-dry mass to carbon mass. Local equations are obtained in mixed dipterocarp forest in Peninsular Malaysia close to code i plots[51] applied for these plots. The other from mixed dipterocarp forest in East Kalimantan[60] were applied to the same forest-type plot in West Kalimantan (code j) and montane forests in West Java (codes d) as well. Equations for heath forest and peat-swamp forest in Central Kalimantan[49] were applied to the same-site plots (code f) with respective forest types ('Lahei-2' as peat-swamp; other two as heath forest). In these local equations, we estimated tree height from stem diameter based on plot-specific inventory data, and obtained plot-specific coefficients of the extended allometry with asymptotic height that link diameter and height[49–51,61].

## Species and forest productivity

We applied the following procedures[20] for estimating per-area forest aboveground woody productivity, i.e., aboveground net productivity by tree growth of surviving stems and ingrowth by recruited stems, that reduce the influence of inter-census interval and among-population variation[20]. We estimate per-area aboveground biomass of species $i$ in a plot, $B_i$ (Mg C ha$^{-1}$), and species relative (i.e., per $B_i$) aboveground woody productivity $p_i$ (yr$^{-1}$) as follows. For a census interval of $T$ (yr), we obtained the aboveground biomass of a species $i$ at the first ($B_{0i}$) and second census ($B_{Ti}$) as the sum of individual tree mass that were alive at each census, divided by the plot area (Mg C ha$^{-1}$). We also obtained the survived fraction of initial biomass (Bsurv$_{0i}$) as the sum of alive tree biomass at the first census that survived until the second census, divided by plot area. Then, using our methods[20], the estimated instantaneous relative aboveground woody productivity of species $i$ is

$$p_i = \ln(B_{Ti}/B\text{surv}_{0i})/T, \qquad (1)$$

and the period-mean aboveground biomass of species $i$ over the two censuses is

$$B_i = (B_{Ti} - B_{0i})/\ln(B_{Ti}/B_{0i}). \qquad (2)$$

The species absolute aboveground woody productivity is $P_i = p_i B_i$ (Mg C ha$^{-1}$ yr$^{-1}$). We also obtain instantaneous relative aboveground woody loss rate (due to tree mortality) of species $i$ by $l_i = \ln(B_{0i}/B\text{surv}_{0i})/T$, and absolute loss rate of $i$ by $L_i = l_i B_i$, that counterbalance $p_i$ and $P_i$, respectivcely[20]. The forest period-mean aboveground biomass $B$ (Mg C ha$^{-1}$) and aboveground woody productivity $P$ (Mg C ha$^{-1}$ yr$^{-1}$) are respectively[20]

$$B = \sum_i B_i, \qquad (3)$$

and

$$P = \sum_i p_i B_i = \sum_i P_i. \qquad (4)$$

Similarly, forest-level rate of aboveground woody loss is $L = \Sigma_i L_i$. In the provided code in the Zenodo repository (https://doi.org/10.5281/zenodo.7668416), we adopted our generalised estimation scheme for relative woody productivity $p$ and relative woody loss rate $l$ by tree mortality of populations for varied inter-census intervals among individual trees[21]. We define forest tree abundance $N_i$ (ha$^{-1}$), or per-area stem count (≥5 cm stem diameter) of population $i$ to be the sum of population period-mean per-area stem counts[61].

In each plot, we selected every species (or morpho-type) with two or more trees that survived through two censuses, and combined all other rarer species (<two surviving trees per plot) within an aggregated multi-species population. Total number of species across 60 plots was 1587, that of per-plot species populations was 3807, and that without aggregated rare populations was 2604. When any species ≥ two surviving trees showed non-positive $p_i$, due to diameter decrease of large tree(s) during the corresponding period, we discarded those populations as in $p_i$-$B_i$ model fitting in Fig. 2a, b (53 out of 2604 populations, i.e., 2.1%), but we included all species with aggregated rare species and with $p_i \le 0$ in forest-level woody productivity and loss rate estimates.

## Productivity model fitting

To fit and quantify power-law models, we applied linear models with log-transformed variables assuming that response variables are

lognormally distributed such as

$$\ln y_j \sim \text{Normal}(\mu_j, \sigma), \quad (5)$$

$$\mu_j = k + b \ln x_j + c\, \text{MAT}_j \quad (6)$$

where $j$ is plot (or species population) identity, $y_j$ is a response variable, $x_j$ and $\text{MAT}_j$ are explanatory variables, $\mu_j$ and $\sigma$ are $j$-specific mean and common residual deviation, and $k$, $b$ and $c$ are model coefficients to be estimated. The logarithmic model of Eq. (6) is converted to the power-law model,

$$y_j = a\, x_j^b \exp(c\, \text{MAT}_j), \quad (7)$$

where $a = \exp(k + \sigma^2/2)$ by adjusting the mean of lognormally distributed $y_j$ from normally distributed $\ln y_j$. In species population level model fitting, we excluded 'aggregated' rare species and records with non-positive $p_i$ as $y_j$, but we included these in forest plot-level analyses. We present conditional $R^2$ values[62] for log-log linear mixed models of species aboveground biomass against species maximum tree size or species abundance (per-area tree count) with plot-specific constant terms in Supplementary Fig. 2. Chemical theory has stimulated that MAT-dependence of turnover rates is related to the inverse of absolute temperature (°C), i.e., $1/(273.15 + \text{MAT})$[22,30]. We used a simpler formulation of Eq. (7) over our limited range of MAT values (0–30 °C), noting the near linear dependence between MAT and $1/(273.15 + \text{MAT})$ ($R^2 = 0.999$).

We carried out data analyses and graphical presentations with R version 4.0.5[63] and Python version 3.9[64]. All data at the level of individual trees and species populations, and the R code used to generate Fig. 4 are provided in the Zenodo repository (https://doi.org/10.5281/zenodo.7668416).

### Null model data of temperature dependence
To test the species response hypothesis (Fig. 1b) separated from community structure difference among plots, we calculated woody productivity of each species population responding to the plot location in terms of the plot-specific power-law constant, $a_{\text{plot}}$, of species productivity ($p_i$) versus biomass ($B_i$), i.e., $p_i \sim a_{\text{plot}} B_i^b$ (cf. Figure 2a), and that frequency distribution of species biomass is the same as the distribution of all $B_i$'s across all the 60 plots. Our species response estimate of forest-level woody productivity for each plot is:

$$P_{\text{specRes}} = a_{\text{plot}} B \left( \sum_{\text{plot}} \sum_i B_i^{1+b} \right) \Big/ \sum_{\text{plot}} B. \quad (8)$$

This estimate assumes the plot species richness is proportional to the plot biomass $B$. In this estimation procedure, we left the aggregated population of rare species unchanged.

To disentangle the contribution of species response hypothesis and community structure hypothesis (Fig. 1), we generated two null-model woody productivity data of 60 plots. To represent the effect of community structure on woody productivity (Fig. 1a), we generated "replaced data" from the observation data, in which we replaced the productivity of a given species $i$, $P_i$, with the value of different species populations with similar species biomass $B_i$. To do this, we performed a weighted random sampling 10,000 times from the species population pool (excluding the target species) of all plots, using the inverse of the square of the difference in $\ln B_i$ between species populations as the weight. We took the mean value of the resampled data as the "replaced" $P_i$ for each species population $i$. The aggregated population of rare species were left unchanged, and excluded from this replacement procedure. The sum of these values was taken as plot-level

productivity estimate representing the community structure hypothesis, $P_{\text{commStr}}$, in Fig. 4b. Supplementary Fig. 6 compares species woody productivity between original and replaced data.

To quantify the contribution of our two hypotheses, we applied regression model of $\ln P_{\text{specRes}}$ (Fig. 4b) and that of $\ln P_{\text{commStr}}$ (Fig. 4c) to the original data with respect to $\ln P$, and compared their residual variances to that of the original data model (Fig. 4a) as: [variance minus the residual variance from projected model estimates] divided by [variance minus the residual variance from original model estimates]. This ratio is 1 if a null model completely explains $\ln P$ as original model does, and is 0 if the null model explains nothing of the variance with respect to $\ln P$.

### Quantifying biases of standard productivity estimates
Standard, 'simple' estimates of aboveground woody productivity, $P\_\text{simple}$ (Mg C ha$^{-1}$ yr$^{-1}$), is the sum of absolute aboveground mass gain by survived and recruited trees of any species, i.e., $P\_\text{simple} = \sum_i (B_{Ti} - B\text{surv}_{0i})/T$, setting the initial mass for recruited trees is the mass at the threshold tree size (5 cm stem diameter here)[17]. We have theoretically shown that this conventional woody productivity estimate of a population $i$, $P\_\text{simple}_i$ is influenced by inter-census interval $T$, latent relative woody productivity $p_i$ (Eq. (1)) and loss rate $l_i$, thus by inter-specific variation of $p_i$ and $l_i$, in such manner that $P\_\text{simple}_i$ of species $i$ with larger $p_i T$ and $(p_i - l_i)T$ are more largely underestimated[20]. It is also influenced by the treatment of unrecorded initial tree mass[19,20].

We obtained standard, simple estimates of forest woody productivity of the 60 plots to evaluate them in comparison to our instantaneous estimates (Eqs. (1)–(4))[20]. The simple productivity estimates were in average 85% in terms of species population woody productivity and 90% of our forest woody productivity, with large inter-specific and among plot variations (Supplementary Fig. 7a, b). At the species-level, simple estimates predicted the exponent of relative productivity versus biomass power-law at $-0.077 \pm 0.009$ in contrast to that of $-0.14 \pm 0.01$ in our estimates (Supplementary Fig. 7c, compared to Fig. 2a). Forest-level model of $P\_\text{simple}$ dependence on standing biomass and temperature was similar to our model of $P$, besides that $P\_\text{simple}$ was smaller than $P$ by about 10% (Supplementary Fig. 7d, compared to Fig. 4a).

### Aboveground net primary productivity
There were monthly records of litter fall collected using litter traps during the corresponding period in 22 out of our 60 plots. We estimated aboveground net primary productivity (i.e., the estimated aboveground woody productivity $P$ plus recorded monthly-sum fine litter fall (i.e., canopy productivity[9]) during the same census period) to validate our analysis for this more inclusive measure of aboveground net productivity (ignoring non-structural organic carbon productivity). Our estimates of forest-level aboveground woody productivity $P$ are log-log linearly correlated with aboveground net primary productivity ($R^2 = 0.88$; Supplementary Fig. 8a). Our model of forest woody productivity as functions of forest biomass and mean annual temperature (Fig. 4a) also apply to forest net primary productivity (Supplementary Fig. 8b).

### Reporting summary
Further information on research design is available in the Nature Portfolio Reporting Summary linked to this article.

## Data availability
Full datasets at the level of all trees and those compiled for species populations have been deposited in the Zenodo repository (https://doi.org/10.5281/zenodo.7668416). The original data of forest plots in Japan is available at https://www.biodic.go.jp/moni1000/findings/data/ (version June 2019). Climate data used in this study, CHELSA

version 2.1, is available at https://chelsa-climate.org/. Source data are provided with this paper.

## Code availability

The code for plot-level analyses is provided in the Zenodo repository (https://doi.org/10.5281/zenodo.7668416).

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

## Acknowledgements

We thank many colleagues and field staffs who carried out plot censuses over decades across the region and updated databases. We acknowledge the Research Center for Ecology and Ethnobiology, National Research and Innovation Agency (BRIN) Indonesia, Forest Research Institute Malaysia (FRIM), Taiwan Forestry Bureau, the Ministry of Science and Technology, Taiwan Forest Research Institute, and the Ministry of the Environment Monitoring Sites 1000 Project, for organizational collaboration. Taiwan plots are part of the ForestGEO network which is a consortium managed by the Smithsonian Tropical Research Institute. This study was funded by Grant-in-Aid for Scientific Research of the Japan Society for the Promotion of Science 18H02504, 21H02248 (T.S.K.), 22H05715 (T.I.K.), 21H05316, 21H02227 (T.H.), and 19K06141 (N.N.).

## Author contributions

T.I.K., D.S., and T.S.K. conceptualized and designed the study. T.I.K., I.F.S., K.N., Z.S.N.H., E.S., T.S.K., S.J.R., T.H., N.N., K.H., S.H.W. and W.C.C. carried out field censuses and data management. T.I.K. and T.S.K. analyzed data. T.I.K., D.S., and T.S.K. draughted the original manuscript, and all authors revised and approved the final version of the manuscript.

## Competing interests

The authors declare no competing interests.
