## [Peer Review File · Nature Communications]

Reviewers' Comments:

Reviewer #1:

Remarks to the Author:

In this study, the authors use forest census-data of 60 1 ha-plots from sub-boreal to tropical biomes, to examine how productivity patterns vary with species richness and size across biomes. This study reveals interesting patterns with a very nice dataset. The results are clearly in line with current findings about the role of forest structure on forest functioning. One central result is the fact that species with similar standing biomass and stature show similar productivity. Decrease in relative productivity with increased biomass is classic in forestry. However, highlighting the consistency of the strength of the relationship across biomes is certainly an interesting point. Although I think that the study is not strong or novel enough to be published in the journal, it should make a great contribution to a more ecology-focused journal, including the top ones. I have several concerns that may be addressed.

Before I read the conclusion, I wonder to what extent the pattern found in the decrease of contribution of small-biomass species in colder biomes can be attributed to the fact that tropical biomes simply harbour more tree species than sub-boreal or temperate ones? Indeed, if species richness is much greater for the same surface unit, one may expect that small-biomass species can be over-represented in comparison to other biomes.

Minor points

- Hydraulic issues under warming climate are frequently reported, mostly in temperate zones, but not only. I would suggest to also consider climate-related index based on water stress during dry season (for instance) in the analyses.
- I understand the focus on relative productivity, but the key point about carbon storage is the total productivity. What patterns can be deciphered when considering absolute productivity vs. stand structure and climate?
- First paragraph of the introduction (l. 50-54): Among factors, environmental ones should be mentioned (soil characteristics and climate). Furthermore I think that referring to 'global scale' productivity is actually misleading here as the effect of species richness on forest productivity has to be examined at the plot/landscape level (as it is the case here), even if extrapolations can be made at the global scale afterwards.

Reviewer #2:

Remarks to the Author:

The authors have written a manuscript addressing tree productivity across a temperature gradient in East Asia based on 60 plots of aboveground biomass data. The main conclusion is that more productive forests have higher numbers of smaller trees, which is an important new finding of global significance. The manuscript is well written, the topic is properly introduced, and the methods are properly chosen, well explained, and allow for scientific replication of this study. I have only two minor comments. In the abstract, you write "using methods that minimise bias." I suggest you omit this sentence or better explain what you mean by it. In the discussion, you should also address several factors and variables that were not used in your study but could further explain the variability in forest productivity, e.g., soil and phosphorus.

Reviewer #3:

Remarks to the Author:

The paper "Forest productivity across a thermal gradient in eastern Asia reflects tree community structure" considers whether variation in productivity across climates reflects adjustment among individual trees and species or changes in community structure analysing data from 60 old-growth forest plots.

The idea is very interesting and provides new insights into the understanding of the relationships between productivity, biodiversity, and ecosystem structure.

Personally, I found the finding that community structure (specifically the presence of small-biomass understory trees that contribute to the community structure and, in turn to the system's productivity) is more important than individual tree growth in terms of productivity very enlightening.

Although I am in favour of the publication of the present manuscript, I have generally found the text extremely wordy (e.g. "individual tree level similarity in relative growth rate (per tree mass)"; "values of species standing aboveground biomass across plots"; etc.) resulting in a low fluency of the manuscript.

Other extremely long sentences do not help the understanding. For instance: "This shared inter-specific trade-off reflects the slower turnover rates for larger sized trees that possess more wood for every unit of energy they capture from sunlight despite the greater energy interception expected among the largest stems".

A major concern is related to the absence of a relationship between moisture and productivity. In fact, I do not find justified that authors "assume MAT represents climatic variation over this range, because annual moisture (i.e., precipitation minus potential evapotranspiration) is high due to monsoon climate, and it does not affect structural or dynamic properties of forests".

Since the annual moisture of the plots in this study (distributed latitudinally from Java to northern Japan) ranges from 275 to 5,518 mm yr⁻¹ 317 (i.e. from quite low humidity to very high), I do not agree with the authors' statement that "all plots are in humid conditions".

Moreover, I do not understand how it is possible that "Annual moisture showed no correlation with MAT or with any forest structural or dynamic variable, in contrast to MAT exhibiting significant correlations to all forest variables (Extended Data Fig. 4). I can't imagine how a site in Hokkaido and another in Java may have the same patterns of MAT and moisture and, more important, how this moisture huge differences "do not affect structural or dynamic properties of forests".

I invite the authors to carefully recheck their data and analysis and better justify whether "employing MAT only to quantify climatic dependence of the 60 plots" is correct.

Furthermore, correlations between MAT and biomass ($R^2=0.14$) and MAT and SR ($R^2=0.09$) are extremely low (Extended Data Fig. 4) and is again quite weird that MAT is so poorly correlated with biomass and SR along a so wide latitudinal gradient and with so huge differences of temperature and moisture. If nothing is wrong with the regression analysis this could represent a big challenge to most of energy based theories of the latitudinal gradient of biodiversity and justify statements such as "Different productivity responses to temperature between populations and ecosystems contradicts theories in which net primary productivity varies with temperature similarly from individuals to ecosystems". On the contrary, if something is missing from the data analysis, correlations may miss some important details.

I, therefore, invite the authors to carefully evaluate the goodness of these very challenging and counterintuitive results before extending these findings to a more general ecological law.

At the same time, it is true that "Species-rich tropical forests generally comprise a greater proportion of short-stature species, and trees of such species (Fig. 3), when compared to lower diversity temperate forests" but the "suggestion that adaptation and acclimation within these old-growth forests result in unexpectedly similar woody productivity for species with similar standing biomass and stature regardless of temperature" must be more supported by results, particularly if independence from temperature is proposed as the reason for equal productivity of species.

It is not clear why authors say that "Forest aboveground biomass was also positively correlated with MAT" if they find an $R^2=0.14$; Fig. 3b), which is very low, but suggest "not with species richness" with an $R^2=0.06$; Extended Data Fig. 4), which is not far from the previous one.

Similarly, it seems not justified why authors state that "average per-species aboveground biomass

decreases over this range ($R^2=0.34$; Fig. 3c) as does individual tree aboveground mass ($R^2=0.10$; Fig. 3d)" whereas 0.10 is an R^2 still low (not far from 0.6 of the value between biomass and species richness where a clear relationship was not highlighted by them) and does not confirm any correlation.

I suggest paying attention to studies that already considered the relationships between canopy height, forest structure and species diversity/biomass at a global scale.

Overall, although I consider the study and ideas within it captivating, I think much more attention should be paid to data analysis and result interpretation by the authors.

Therefore, I recommend a comprehensive revision and resubmission of the manuscript.

Reviewers' comments and authors' responses

Reviewer #1 (Remarks to the Author):

In this study, the authors use forest census-data of 60 1 ha-plots from sub-boreal to tropical biomes, to examine how productivity patterns vary with species richness and size across biomes.

This study reveals interesting patterns with a very nice dataset. The results are clearly in line with current findings about the role of forest structure on forest functioning. One central result is the fact that species with similar standing biomass and stature show similar productivity. Decrease in relative productivity with increased biomass is classic in forestry. However, highlighting the consistency of the strength of the relationship across biomes is certainly an interesting point.

Although I think that the study is not strong or novel enough to be published in the journal, it should make a great contribution to a more ecology-focused journal, including the top ones. I have several concerns that may be addressed.

Re: Thank you for your review and comments. We are not unbiased but believe in contrast that the research is both strong and novel (indeed surprising as the other reviewers recognize) and should interest a broad audience if we can make it sufficiently accessible. We developed our paper to appeal to a broad cross-section of readers (specialists and otherwise) in earth and environmental sciences.

Before I read the conclusion, I wonder to what extent the pattern found in the decrease of contribution of small-biomass species in colder biomes can be attributed to the fact that tropical biomes simply harbour more tree species than sub-boreal or temperate ones? Indeed, if species richness is much greater for the same surface unit, one may expect that small-biomass species can be over-represented in comparison to other biomes.

Re: Your concern is to evaluate the influence that high species richness divided by stand biomass simply results in low per-species biomass. We agree it is also the case of our data. Actually, we found that community structure in terms of species biomass

distribution was different along mean temperature (Fig. 2c) besides species richness difference. In our revision, we revised to make this point clearer, by adding new Fig. 3d (standard deviation of species biomass negatively correlated with temperature). Also in Introduction, we mentioned that the proportion of short-stature species is much higher in tropical forests than in temperate forests (with citations).

Minor points

- Hydraulic issues under warming climate are frequently reported, mostly in temperate zones, but not only. I would suggest to also consider climate-related index based on water stress during dry season (for instance) in the analyses.

Re: Thank you for pointing this out. We extended climate variables, with monthly precipitation and potential evapotranspiration (in new Extended Data Table 2). Our sites are mostly ‘perhumid’ and some ‘humid’ category by Thornthwaite (1948). We found no contribution of either AP or AP–PET on the Fig. 4 model ($\log(\text{productivity}) \sim \log(\text{biomass}) + \text{MAT}$). We also did not find any contribution from other variables, such as precipitation amount during the dry season (bio14 and bio17 in CHELSA).

- I understand the focus on relative productivity, but the key point about carbon storage is the total productivity. What patterns can be deciphered when considering absolute productivity vs. stand structure and climate?

Re: We are not quite sure if we understand: there are clearly different measures of productivity and of carbon flows and storage. Storage is not itself a de-facto function of production as we can imagine a forest of similar biomass turning over more or less quickly (higher or lower measures of stand productivity). In any case, we presented relative productivity vs. biomass at species level (Fig. 2a), and also showed absolute productivity vs. biomass at stand level (Fig. 4). We show the difference in Fig. 1. We also added species level absolute productivity vs. biomass relationship in new Extended Data Fig. 4. We have also made some minor edits that should make this clearer.

- First paragraph of the introduction (l. 50-54): Among factors, environmental ones should be mentioned (soil characteristics and climate).

Re: We added, and returned to them in the Discussion.

Furthermore I think that referring to 'global scale' productivity is actually misleading here as the effect of species richness on forest productivity has to be examined at the plot/landscape level (as it is the case here), even if extrapolations can be made at the global scale afterwards.

Re: Ok, right thanks, we have had a lot of discussion on such labels (various views from different authors and trial readers)—we need to find the right degree of precision and clarity without becoming awkward and cumbersome, there is some subjectivity in making these choices. We have adjusted to “stand level” to represent the plot scale.

Reviewer #2 (Remarks to the Author):

The authors have written a manuscript addressing tree productivity across a temperature gradient in East Asia based on 60 plots of aboveground biomass data. The main conclusion is that more productive forests have higher numbers of smaller trees, which is an important new finding of global significance. The manuscript is well written, the topic is properly introduced, and the methods are properly chosen, well explained, and allow for scientific replication of this study.

Re: Thank you very much for reviewing our manuscript. We appreciate your positive validation.

I have only two minor comments. In the abstract, you write "using methods that minimise bias." I suggest you omit this sentence or better explain what you mean by it.

Re: We removed as it was too short. (We would state “using new estimation methods that minimise biases that arise from comparing heterogeneous populations”, which may be too long in Abstract.) We rewrote the second paragraph of Introduction that explains the methodological issue in relation to the significance of species-based productivity estimation.

In the discussion, you should also address several factors and variables that were not used in your study but could further explain the variability in forest productivity, e.g., soil and phosphorus.

Re: Thank you. We revised to explain that the present plots only represent a thermal gradient, and added the other climate indices in the new Extended Data Table 2. We mentioned further needs of evaluating productivity dependence on water and nutrients in the Discussion. There were two heath forests and one peat-swamp forest in central Borneo, two plots on limestone in southern Taiwan and in an Ogasawara Islands, and some difference in humidity indices—our tests did not reveal any influence on model and results.

Reviewer #3 (Remarks to the Author):

The paper "Forest productivity across a thermal gradient in eastern Asia reflects tree community structure" considers whether variation in productivity across climates reflects adjustment among individual trees and species or changes in community structure analysing data from 60 old-growth forest plots.

The idea is very interesting and provides new insights into the understanding of the relationships between productivity, biodiversity, and ecosystem structure.

Personally, I found the finding that community structure (specifically the presence of small-biomass understory trees that contribute to the community structure and, in turn to the system's productivity) is more important than individual tree growth in terms of productivity very enlightening.

Re: Thank you very much for your review comments and suggestions. We appreciate your positive validation, and detailed suggestions. We revised to make our points clearer based on your comments.

Although I am in favour of the publication of the present manuscript, I have generally found the text extremely wordy (e.g. "individual tree level similarity in relative growth rate (per tree mass)"; "values of species standing aboveground biomass across plots"; etc.) resulting in a low fluency of the manuscript.

Re: Thank you for this point. Much of the complexity arises from the need to be specific about what populations, variables and relationships we are dealing with. We fully checked and rephrased complex sentences to be clearer without losing the necessary

precision of language. The first phrase is revised as “relative growth rates of individual trees against tree mass were similar across plots and biomes”. The second example is revised as “per-area species population biomass ...”.

Other extremely long sentences do not help the understanding. For instance: "This shared inter-specific trade-off reflects the slower turnover rates for larger sized trees that possess more wood for every unit of energy they capture from sunlight despite the greater energy interception expected among the largest stems".

Re: We checked and revised long sentences. We separated this sentence into three as “This inter-specific trade-off reflects the slower turnover of larger compared to smaller tree species^{6,21}. While larger trees may capture more solar energy per unit of foliage due to their greater stature²⁸ they also possess more overall biomass per unit of leaf mass compared to smaller trees²⁹. Thereby, taller trees and large-biomass species generally show lower relative productivity than smaller, shorter tree species that also tend to have lower biomass per unit area.”

A major concern is related to the absence of a relationship between moisture and productivity. In fact, I do not find justified that authors “assume MAT represents climatic variation over this range, because annual moisture (i.e., precipitation minus potential evapotranspiration) is high due to monsoon climate, and it does not affect structural or dynamic properties of forests”.

Since the annual moisture of the plots in this study (distributed latitudinally from Java to northern Japan) ranges from 275 to 5,518 mm yr⁻¹ 317 (i.e. from quite low humidity to very high), I do not agree with the authors' statement that "all plots are in humid conditions".

Moreover, I do not understand how it is possible that "Annual moisture showed no correlation with MAT or with any forest structural or dynamic variable, in contrast to MAT exhibiting significant correlations to all forest variables (Extended Data Fig. 4). I can't imagine how a site in Hokkaido and another in Java may have the same patterns of MAT and moisture and, more important, how this moisture huge differences "do not affect structural or dynamic properties of forests".

Re: Thank you. We had explored these relationships previously and have now added more details by way of explanation. We have added, considered and explained

information on climate and indices on moisture and humidity (new Extended Data Table 2), and revised to address moisture in the present dataset. For example, in the revised Extended Data Fig. 5 we examine the pairwise relationship between AP, AP–PET and forest variables. Though AP is generally higher in low latitude with high MAT, high temperature brings about higher saturated water vapor pressure. To address relative moisture on the basis of monthly precipitation and potential evapotranspiration, we show that sites are mostly in ‘perhumid’ (Thornthwaite Type A), or in ‘humid’ (Type B) climate. We also show through formal tests that neither AP nor AP–PET provide additional influence on the forest productivity model in Fig. 4a.

I invite the authors to carefully recheck their data and analysis and better justify whether "employing MAT only to quantify climatic dependence of the 60 plots" is correct.

Re: We have checked. We have added the TMI (Thornthwaite Moisture Index) based on monthly precipitation, temperature and evapotranspiration, in the new Extended Data Table 2.

Furthermore, correlations between MAT and biomass ($R^2=0.14$) and MAT and SR ($R^2=0.09$) are extremely low (Extended Data Fig. 4) and is again quite weird that MAT is so poorly correlated with biomass and SR along a so wide latitudinal gradient and with so huge differences of temperature and moisture. If nothing is wrong with the regression analysis this could represent a big challenge to most of energy based theories of the latitudinal gradient of biodiversity and justify statements such as "Different productivity responses to temperature between populations and ecosystems contradicts theories in which net primary productivity varies with temperature similarly from individuals to ecosystems". On the contrary, if something is missing from the data analysis, correlations may miss some important details.

Re: Thank you for your detailed evaluation. We have checked and now provide more detail to share on these checks. In our pairwise semi-log regression, species richness SR is more strongly correlated to MAT (with $R^2 = 0.64$; Fig. 3a), than is aboveground biomass against MAT ($R^2 = 0.14$; Fig. 3b). Meantime, SR and biomass are only weakly correlated ($R^2 = 0.09$, in Extended Data Fig. 5). Revised Extended Data Fig. 5 shows apparent patterns for AP. We evaluated the forest productivity model (Fig. 4a), finding that AP provides no detectable influence.

I, therefore, invite the authors to carefully evaluate the goodness of these very challenging and counterintuitive results before extending these findings to a more general ecological law.

At the same time, it is true that "Species-rich tropical forests generally comprise a greater proportion of short-stature species, and trees of such species (Fig. 3), when compared to lower diversity temperate forests" but the "suggestion that adaptation and acclimation within these old-growth forests result in unexpectedly similar woody productivity for species with similar standing biomass and stature regardless of temperature" must be more supported by results, particularly if independence from temperature is proposed as the reason for equal productivity of species.

Re: Thank you. It is important to us to keep the presentation clear and accessible, and hear it appears we needed to revise and improve. We revised our wording here to avoid confusion. In our present results, we only tested by null-models, thus to test the influence of community structure beyond East Asian MAT gradient, further test with varied community structure independent of MAT (e.g. monodominant tropical old-growth forests in particular habitat) will be required.

It is not clear why authors say that "Forest aboveground biomass was also positively correlated with MAT" if they find an $R^2=0.14$; Fig. 3b), which is very low, but suggest "not with species richness" with an $R^2=0.06$; Extended Data Fig. 4), which is not far from the previous one.

Re: Good point. We rephrased this section for clarity. We carried out stepwise multiple correlation analysis of productivity model from the set of these variables:
 $\log(\text{productivity}) \sim \log(\text{biomass}) + \log(\text{SR}) + \log(\text{abundance}) + \text{MAT} + \text{AP}$
(by 'lm' expression on R) and the selected model included just $\log(\text{biomass})$ and MAT because of the collinearity between SR and MAT.

Similarly, it seems not justified why authors state that "average per-species aboveground biomass decreases over this range ($R^2=0.34$; Fig. 3c) as does individual tree aboveground mass ($R^2=0.10$; Fig. 3d)" whereas 0.10 is an R^2 still low (not far from 0.6 of the value between biomass and species richness where a clear relationship

was not highlighted by them) and does not confirm any correlation.

Re: Ok, thanks. We have rephrased to emphasise the weak productivity vs. biomass correlation ($R^2 = 0.14$) compared to the stronger productivity vs. SR correlation ($R^2 = 0.64$), which brought about the marked *negative* correlation between species mean biomass and MAT ($R^2 = 0.34$). In contrast, mean tree mass vs. MAT in old Fig. 3d ($R^2 = 0.10$). We decided to remove the old Fig. 3d, but added a new Fig. 3d that indicates standard deviation of species biomass per stand decreases with MAT.

I suggest paying attention to studies that already considered the relationships between canopy height, forest structure and species diversity/biomass at a global scale.

Re: Thank you, we had been trying to be brief and succinct. We have now developed these themes further in the discussion.

Overall, although I consider the study and ideas within it captivating, I think much more attention should be paid to data analysis and result interpretation by the authors.

Therefore, I recommend a comprehensive revision and resubmission of the manuscript.

Re: We have made a comprehensive revision and now resubmit. As the reviewers note the ideas are surprising and stimulating and we understand that our analyses may not be able to answer every concern and quibble while also being clear and succinct. But indeed we need to find the right balance ... hopefully we have done so. Thank you again for the detailed appraisal it has forced us to be precise, clear and specific.

Reviewers' Comments:

Reviewer #2:

Remarks to the Author:

The authors have addressed my two minor concerns, and I am satisfied with their response.

Reviewer #3:

Remarks to the Author:

I thank the authors for having considered and addressed my comments. From my side, the paper can now be published.

Thanks

Reviewer #4:

Remarks to the Author:

The review done by the authors correctly addressed all the concerns raised by the reviewers, and I suggest publication of the manuscript in its present form

Nature Communications NCOMMS-22-31498B

We thank the reviewers for their careful review and evaluation on our paper.

Yours sincerely,

Tetsuo Kohyama with all co-authors